# LONG BERT FOR BANKRUPTCY PREDICTION

## ABSTRACT

Most bankruptcy risk prediction models use numerical data such as financial statements, financial ratios, or stock market variables to predict the risk of a company going into bankruptcy. However, these models do not take advantage of the vast amount of textual information available. The few projects that work with textual information use short texts such as tweets and news or are limited to analyzing data from public companies. Our research focuses on predicting the bankruptcy risk using the long text sequences of the annexes from the Annual Accounts. We propose a BERT-based model, which can predict the risk of a company going bankrupt even if there is no explicit information about the risk in the long-textual information. Here we showed that we can process parallel segments of a document using BERT and then integrate them for a unified prediction. Using a dataset of 20,000 annexes from the Annual Accounts of non-financial companies from Luxembourg to train and validate our model. We tried different models and two of them get a validation precision for predicting a risky company of approximately 73% and can be used depending on how long the documents are. The model can clearly learn about risk information from unstructured and diverse long textual information with high precision. This is our first step towards an integrated learning model that considers also numerical and non-financial data. Our proposed architecture can be used in other domains where long text needs to be processed for different Natural Language Processing tasks.

## 1 INTRODUCTION

Forecasting bankruptcies plays a pivotal role in fostering economic growth, benefiting businesses, creditors, investors, and even government entities. The presence of effective early warning systems can substantially diminish bankruptcy risks and pinpoint vulnerable aspects that require a company's attention. Despite the increasing practice of governments and companies disclosing their financial reports, there remains an underutilization of the vast publicly available textual data.

Most of the current financial models are focused on the use of numerical information as financial ratios to make their predictions and only a few of them are using textual information. Most of these text-based models use only short text for making predictions such as financial news and tweets (Qu et al., 2019; Clement et al., 2020).

The study conducted by Mai et al. (2019) is one of few that uses financial disclosures to predict the risk of bankruptcy. However, their dataset is limited in scope as they use the Annual Filings from *public U.S. corporations* which requires *including risk analysis* in the first part of the document, therefore they just need to process a short text. The primary constraint restricting the utilization of extensive textual data sources, such as the complete Annual Reports, is the limitation imposed by the maximum amount of input text that can be processed by state-of-the-art Natural Language Processing (NLP) models like BERT (Devlin et al., 2018).

In contrast to Mai et al. (2019), we are using Annual Accounts from the Luxembourg Business Registry LBR (2023) which correspond to companies from different industries and sizes and even languages(French, German, and English). In these legal annexes the risk is not always explicitly written, so the model should analyze the entire content to detect it.

LBR Annual Accounts consist of the Financial Statements (Balance Sheet and Profit and Loss Statements) and the Legal Annexes or Appendixes. These annexes provide additional information to the Financial Statements using natural language. Although the Financial Statements must follow a spe-

cific template, there is no set layout for the legal annexes [1]. In this sense, our information is less uniform and more variable. We can have annexes that are as short as a single page or as long as 15 pages or more. This requires having models that are capable of receiving long strings.

Our research contribution encompasses two primary objectives. Firstly, we endeavor to enhance our capacity to efficiently process extensive textual data. Secondly, we aspire to build a model proficient in evaluating the bankruptcy risk associated with any company, utilizing the substantial textual content available within their Annual Accounts' annexes. Our specific emphasis is on scenarios where the risk description may not inherently reveal itself. Providing the model with the whole information, it can learn by itself the features required for doing bankruptcy forecasting.

The model that we propose is based on the BERT model, which is one of the most important models in NLP. Our proposed model combines the power of BERT for analyzing short sequences of texts with an Integration network, which can be a Long Short-Term Memory (LSTM) network or a concatenation layer followed by dense layers, to analyze BERT's outputs sequentially.

Our preliminary results show a precision risk prediction of approximately 73%. Although our results may not surpass the performance of models based solely on numbers, we have very good infer that the text provides us with certain information about the level of risk of a company, even if it is not explicitly stated. Furthermore, this model can be applied to any domain where processing long text is required.

## 2 Related work

Most of the studies in finance use numerical data as financial ratios to predict the risk of bankruptcy (Qu et al., 2019; Clement et al., 2020). However, there are some studies that use textual information, but most of them are related to short text as input such as tweets or financial news (Sawhney et al., 2021; Sandy et al., 2022; Gao et al., 2018). In contrast, our objective is to analyze financial reports to predict the risk of bankruptcy. This leads us to use long text as input for our model

The first work on using financial disclosures to predict bankruptcy risk was done by Mai et al. (2019). They used financial disclosures of U.S. Corporations to predict the risk of bankruptcy using their publicly available 10-K annual filings. the Security Exchange Commission (SEC) mandated firms to include the risk information into a *"risk factor"* section in the 10-K form starting in 2005 (Campbell et al., 2014; Huang AAllen, 2020).

In contrast to Mai et al. (2019), our study uses a publicly available dataset of Annexes of Annual Accounts from Luxembourg Business Registers (LBR). By law, not only public companies but also SMEs are required to produce annual accounts. These filings can be presented in English, French, or German. Those annexes can have one or many pages. There is no regulation about the content in the annexes, therefore, the information in the notes or annexes can be very limited (OpenLux (2021)).

For processing textual information using Deep Neural Networks, the current leading technologies are the models based on the Transformer Architecture (Vaswani et al., 2017). These models replaced the Long Short-Term Memory Networks (LSTM) (Kang et al., 2020). Transformers use *attention mechanism* to do parallel processing and avoid the vanishing gradient problem.

Based on Transformers, Devlin et al. (2018) presented the BERT model. BERT is composed of one embedding layer and at least 12 attention layers (BERT Base). It is designed to deal with the ambiguity of language using the surrounding text to define the context. BERT has become one of the most important base models for researchers in NLP. This pre-trained model can understand general language and can be easily trained to understand domain-specific terms without requiring high computational resources or huge amounts of data. In HuggingFace (2023) there are more than 9,000 BERT-based models for NLP in different domains, languages, and downstream tasks such as Text Classification, Translation, Question-Answering, etc.

Moreover, the primary limitation of BERT-based models is their maximum input size, which is 512 tokens. A token can be defined as a word or word piece that has a numerical representation and meaning. The tokenizer is the component of the model which has a dictionary of known words or

---

[1]https://guichet.public.lu/en/entreprises/gestion-juridique-comptabilite/comptable/enregistrement/methodes-etablissement-comptes-annuels.html

word pieces and decomposes the input text into these known entries. Each entry has a vector of 768 features that represent the token's embeddings.

Our research is focused on documents with several pages, which is different from most of the other studies. With the same goal and same data as Mai et al. (2019), Kim & Yoon (2021) used a BERT-base model, limiting the number of input tokens to 512 and merging the textual information with five financial variables calculated from the stock market. Their BERT model is used to domain-adapt the financial vocabulary and then they use the embeddings for three different classification models. These models are (A) Hazard model, which uses logistic regression, (B) k-Nearest Neighbors (k-NN) and (C) Support Vector Machine (SVM). These models are not DL models and use distance similarity to make predictions.

In consequence, by default, a BERT instance can not be fed with long textual information like the financial annexes. To solve this problem, likewise Zhang et al. (2020) we are using a Long Short-Term Memory (LSTM) network. In contrast to them, instead of including an LSTM layer after each attention layer, we include an LSTM layer after BERT or we concatenate BERT outputs, having some dense layers in between. All the groups are processed in parallel by BERT and then analyzed in sequence by the LSTM network or a concatenation layer. Hence, while BERT generates a general understanding of each part of the document (summarizing), the LSTM or concatenation network analyzes these parts and provides a single prediction for the whole document.

To have a basic idea of the performance of the financial ratios for predicting the risk of bankruptcy, in 2022, (removed for bling review) analyzed the risk of bankruptcy using the Financial information from the Financial Statements from the LBR. Using the most typical financial ratios for one-year bankruptcy prediction they achieved an AUC of 75% for Logistic Regression, 79% for Random Forest, and 87% for LightGBM.

## 3 DATASET

The dataset consists of the Annexes of 20,000 Annual Accounts from the Luxembourg Business Registers (LBR), which are publicly available for download[2]. This dataset contains Small and Medium Companies (SME's) from 16 different industries (Excluding financial institutions, to avoid bias in the reporting). Our dataset contains documents mainly in French ($\approx 81\%$), but also in English ($\approx 11\%$) and German ($\approx 8\%$). More information about the dataset can be found in Appendix A.

For analyzing the risk of bankruptcy in this dataset ("Risky" or "Not Risky"), we labeled the documents based on the status of the company of the last filled report or published court order. As a result, $\approx 20\%$ of the data is *Risky* and the rest is *Not Risky*. The data is imbalanced as well at the language level, keeping similar distributions Appendix A.

## 4 METHODOLOGY AND PROPOSITION

For working with long text we proposed two main architectures composed of a parallel BERT processing and then an integration that can be implemented with an LSTM network or a set of dense and concatenation layers.

Our proposed model consists of three phases: A) Text pre-processing, B) Segment Analysis, and C) Integrated prediction. Both approaches share phase A and the first part of phase B Figure 1.

For Segment Analysis and Integrated Prediction, we have different approaches and with some combinations of these two phases, we create different models to test and select the best one.

### 4.1 TEXT PREPROCESSING

Figure 2 shows the process of our first phrase. This phase is composed of three steps: Cleaning, Tokenization, and Segmentation.

At first, the data is cleaned. We remove e-mails and multiple blank spaces. Then, using regular expressions, we remove dates, apostrophes, dashes, enumerators, and noisy characters. Enumerators

---

[2]https://www.lbr.lu/mjrcs/jsp/webapp/static/mjrcs/en/mjrcs/legal.html?pageTitle= footer.legalaspect

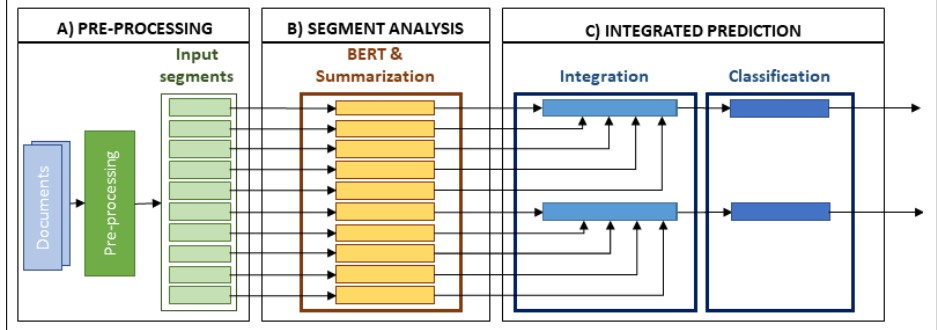

Figure 1: The analyzed models are composed of three phases. A) Pre-processing, where the data is prepared for parallel long processing of each document divided into segments. B) Segment Analysis, where each segment is processed by BERT independently. C) Integrated prediction, where the different segments are integrated to make a prediction using the whole data.

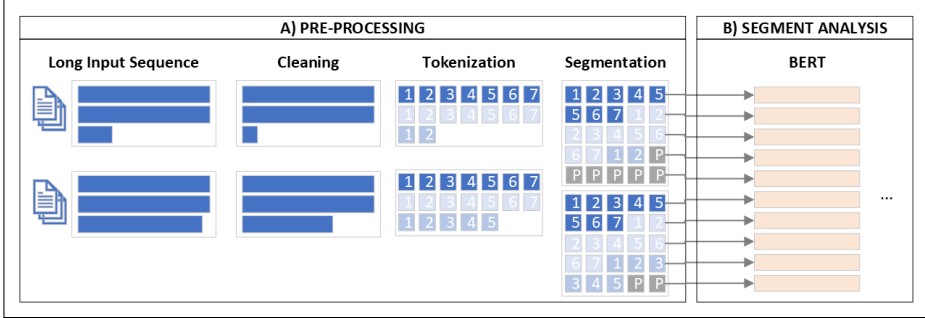

Figure 2: During the first phase, the data is cleaned, tokenized, and segmented. During segmentation, we create segments of 512 tokens. The last part of each segment is copied to the beginning of the next segment to preserve context. Short segments are completed with *[PAD] tokens.*

are common in reports and can be found as Arabic numbers (1.,1.5., etc), Roman numbers (I., IV., etc), and letters (a., b., etc) or a combination of them. As shown in these examples, enumerators can represent a hierarchy and can be delimited with dots, dashes, colons, parentheses, etc.).

After cleaning, we start the tokenization, where each word is divided into one more tokens. The tokenization is done by the default BERT multilingual tokenizer (bert-base-multilingual-cased) [3].

Segmentation is the last step of this phase. Here, we adapt our tokenized data to the input restrictions of the BERT model. The integrity of the document is preserved during the whole process, but until this phase, we do not split the input into independent instances. For this reason, the shuffling and selection of train, test, and validation data are done before the pre-processing. BERT model only accepts 510 tokens for classification tasks (considering the special tokens *[CLS]* and *[SEP]*). After we divide the whole text into segments, each segment is going to be processed by BERT independently. In consequence, context outside the segment is lost. As the meaning of the words depends on the context, we are copying the last $\delta$ tokens of each segment to the next one. The number of segments per document can be estimated with Equation 1, where $\tau$ is the number of tokens generated per document and $BERT_{input}$ is the number of available input tokens (510).

$$\approx \delta_{doc_i} = \lceil (\tau_{doc_i} - \delta)/(BERT_{input} - \delta) \rceil \tag{1}$$

To proceed with the pre-processing phase, we need to define the maximum number of segments per document ($\kappa$), which is mandatory to uniform the BERT output. The value of $\kappa$ is determined according to our dataset. In our dataset, we defined $\kappa = 10$, which covers most of the 80% of the

---

[3]https://huggingface.co/bert-base-multilingual-cased

Table 1: Different values for $\kappa$.

| $\kappa$ | 5 | 10 | 15 | 20 | 25 |
|---|---|---|---|---|---|
| Trimmed documents | 129,532 (59.5%) | 38,225 (82.4%) | 19,092 (91.2%) | 8,596 (96.1%) | 3,703 (98.3%) |

documents without being trimmed. Is important to remind that if the value of $\kappa$ increases, the size and the sparsity of the data also increase. Hence, it could affect the model's performance.

For each document we create $\kappa$ segments, if the number of required segments of the documents is bigger than the maximum number of segments, we trim the first part of the text to fit into the maximum number of segments: $\delta_{doc_i} = min(\approx \delta_{doc_i}, \kappa)$.

If the number of tokens in the last segment of a document is less than $BERT_{input}$, we complete it with *[PAD]* tokens (left padding). If the number of segments is less than $\kappa$, we complete with segments full of *[PAD]* tokens (bottom padding).

### 4.2 SEGMENT ANALYSIS

In segment analysis, the segments are fed into parallel BERT instances. (a) To have a direct connection to the integration phase and (b) to include after BERT, dense layers to reduce the BERT's output dimension (768). The *[CLS]* output vector of BERT is directly fed to the next Integrated prediction phase (first approach) or can be fed into two consecutive dense layers to reduce the dimensionality before feeding into the next phase (second approach). Detailed graphs can be found in Appendix A

### 4.3 INTEGRATED PREDICTION

In this phase, we analyze together the different segments' representations that belong to the same document and then optionally have an intermediate dense layer to reduce the number of dimensions of the integration layer and finally use a dense layer for the classification task. The size of the last output layer of this part of the model is the number of our classes (2).

Similarly to the previous phase, in this phase, we propose three different approaches. The first approach consists of using the last classification layer just after the LSTM network. For the second approach, we include two intermediate dense layers between the LSTM network and the last classification layer. Finally in the third approach, instead of having a LSTM layer, we concatenate the outputs of the previous phase and then reduce the dimensionality with two intermediate dense layers (similar to the second approach). Detailed graphs can be found in Appendix A

### 4.4 MODELS' CONFIGURATIONS

Based on the different approaches, we designed six different integrated models to check which one is the best performed. Each model has the associated parameters for the different dense layers. In the case of the LSTM, we are reporting the different values of the corresponding number of hidden layers ($\phi$) in subsection 5.1. This parameter has a significant impact on the size and performance of the model. The resulting model's architecture is shown in Figure 3

## 5 EXPERIMENTS AND RESULTS

We defined the size of the dense layer according to its position and function in the network. Table 2 shows the formulas per each $\lambda_x$. Most of the values are the power of 2 or an even value, having the second dense layer as half of the previous one. The value of $\lambda_1$ is defined by the closest lower power of two of the size of BERT's output (768). $\lambda_2$ is the half of the previous layer. We fixed the value of $\lambda_3$ big enough to then do the summarization. We defined this value as $\approx 2 * \phi$. $\lambda_4$ also is defined as the half size of the previous dense layer.

For the last two models, which do not depend on LSTM, we use the size of the concatenated sequence as the reference. $\lambda_5$ is half of that. $\lambda_3$ also depends on the size of the concatenated sequence, trying to get the number before the lower power of 2 of the sequence. Finally, $\lambda_4$ is defined as half of the previous dense layer.

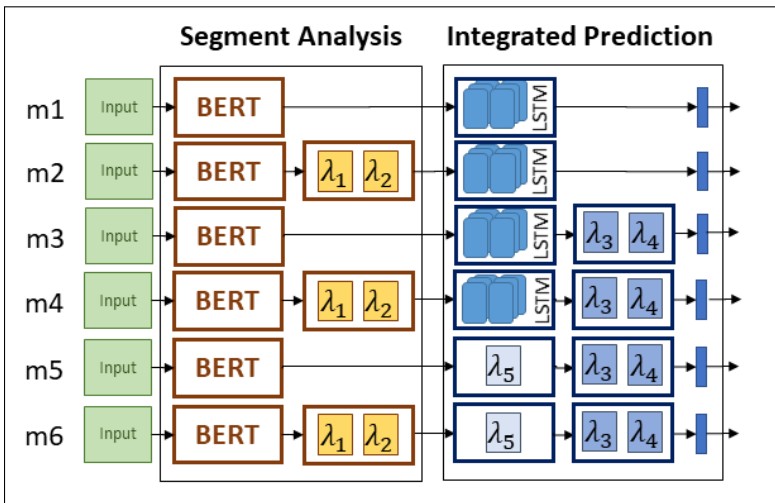

Figure 3: The resulting models are the mixture of the different approaches for each phase. The figure shows the parameters of the dense layers $\lambda_x$, and it represents the output size of the corresponding dense layer.

Table 2: Values for $\lambda_1, \lambda_2, \lambda_3, \lambda_4, \lambda_5$ per model

| Models | Parameter | Value | Explanation |
|---|---|---|---|
| m2,m3 | $\lambda_1$ | 512 | $\lfloor \log_2 768 \rfloor$ |
| m2,m3 | $\lambda_2$ | 256 | Half of the previous dense layer |
| m3,m4,m5,m6 | $\lambda_3$ | 512 | $\lceil \log_2(2 * max(\phi)) \rceil$, having $max(\phi) = 200$ |
| m3,m4 | $\lambda_4$ | 256 | Half of the previous dense layer |
| m5 | $\lambda_5$ | variable | $768*\kappa/2$, half of the concatenated sequence |
| m6 | $\lambda_5$ | variable | $256*\kappa/2$, half of the concatenated sequence |
| m5 | $\lambda_3$ | variable | $2^{(\lfloor \log_2(768*\kappa) \rfloor - 1)}$ |
| m6 | $\lambda_3$ | variable | $2^{(\lfloor \log_2(256*\kappa) \rfloor - 1)}$ |
| m5,m6 | $\lambda_4$ | variable | Half of the previous dense layer |

## 5.1 TEST CASES

For the BERT models, we use the following hyperparameters: $dropout : 20\%$, freezing the first ten first encoder layers of BERT The general model parameters: $learning rate = 1e^{-5}$, $optimizer :$ $AdamW$ with $weight decay = 1e^{-2}$ and $batch\_size = 5$, $epochs = 4$. We freeze 10 out of 12 encoder layers to reduce the trainable model's size.

The test cases are going to depend on the parameters already defined in Table 2 and the value of the number of hidden layers for LSTM networks ($\phi$). This parameter can have the following values [30,50,80,120,200]. The codification of the test case for showing later in the results are following the pattern: $tc\_m_\omega\_h_\phi$, having $\omega$ as the model number. For models 4 and 5, as there is no parameter $\phi$, the pattern is $tc\_m_\omega$.

Additionally, with the best-performed model, we do a sequence length analysis, where we tried sequence lengths from 5 to 20. We decided to remove the test of 25 because the size of some models did not fit in the GPU. We are codifying these test cases like $tc\_m_\omega\_s_\kappa$.

For the analysis of the best model, we are going to show the Accuracy, F1 score, precision, and recall. We are using Accuracy and F1 score calculated with the sklearn toolkit using weighted methods. For comparing and choosing the best model, we are using precision, because we prefer to have more *Risky* companies, predicted as *Risky*.

## 5.2 RESULTS

We conducted all experiments on a Tesla V100-SXM2-16GB GPU with 200GB RAM.

Table 3: Results of Risk Precision of the models with respect to the different number of hidden layers (h).

|     | h30    | h50    | h80    | h120   | h200   |
| --- | ------ | ------ | ------ | ------ | ------ |
| m1  | 0.5187 | 0.5204 | 0.5126 | 0.4638 | 0.6427 |
| m2  | 0.0139 | 0.0165 | 0.0652 | 0.6083 | 0.6510 |
| m3  | 0.5892 | 0.6205 | 0.6171 | 0.5256 | 0.6405 |
| m4  | 0.5839 | 0.5552 | 0.7330 | 0.4734 | 0.5230 |
| m5  | 0.6021 | 0.6021 | 0.6021 | 0.6021 | 0.6021 |
| m6  | 0.4856 | 0.4856 | 0.4856 | 0.4856 | 0.4856 |

Table 3 shows the summary Risk Precision metric for all the models with respect to a variable number of hidden layers (h). A detailed report of other metrics can be found in Appendix B.

From the results, we can conclude that the model m1 is the simplest model with an LSTM network. The maximum precision is given by an intermediate configuration of 200 hidden layers. Most of the configurations perform similarly.

For the model m2, we reduce the BERT's dimensionality with intermediate dense layers before the LSTM network. Is clear that the precision is really poor for the first three configurations, and gets a significant improvement from 120 hidden layers.

Model m3 reduces the dimensionality of the LSTM network before the classification layer. For most of the configurations, the performance is similar. Having 200 hidden layers as the best-performed model.

The model m4 is the most complex model, where we reduce the dimensionality for BERT's output and LSTM's output with intermediate dense layers. Most of the configurations have performance above 50%. The best-performed configuration reaches around 70% of precision for an intermediate model with 80 hidden layers.

As previously mentioned, for models 5 and 6 we do not change the number of hidden layers, that is why we execute these experiments once to compare with the rest of the models. Results show that a direct concatenation from BERT's output performs better than a previous dimensionality reduction.

Figure 4 shows the results of the precision performance for all the models and the number of configurations of hidden layers. The best-performed model of all is the m4 model with 80 LSTM hidden layers. Is important to see that concatenation-based models do not have a bad performance. Even model m5, which concatenates directly the BERT's output, has very good precision, similar to most of the other model's configurations.

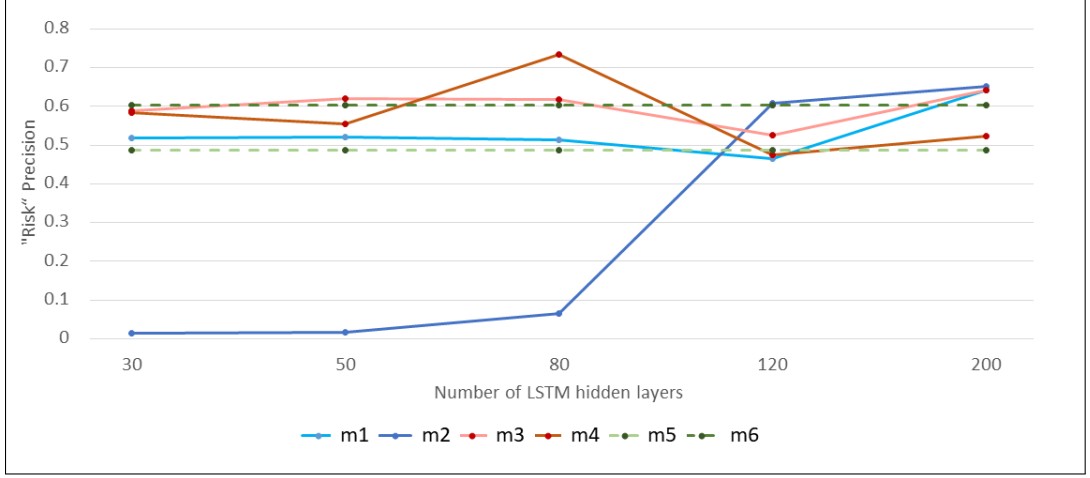

Figure 4: Comparison of Results of the different models with respect to the number of hidden layers. Model 5 and model 6 do not use LSTM, therefore we are showing the same result for all the values in the x-axis.

Table 4: Results of Risk Precision of the models with respect to different sequence lengths (s).

|  | s5 | s10 | s15 | s20 |
|---|---|---|---|---|
| m1 | 0.6706 | 0.5995 | 0.0007 | 0.0007 |
| m2 | 0.5752 | 0.0007 | 0.0007 | 0.0007 |
| m3 | 0.6244 | 0.2663 | 0.0007 | 0.0007 |
| m4 | 0.4657 | 0.6402 | 0.5745 | 0.0007 |
| m5 | 0.5991 | 0.5340 | 0.6624 | 0.7205 |
| m6 | 0.8017 | 0.4682 | 0.6982 | 0.6661 |

Table 4 shows the summary Risk Precision metric for all the models with respect to a variable number of input segments (s). A detailed report of other metrics can be found in Appendix B.

Executing the best-performed model with different sequence lengths, we can see the behavior of each model with respect to the length of the input document. For this comparison, we only executed each model one epoch, to see how difficult could be for the model to start predicting the unbalanced class. As seen in Table 4 some models are just predicting only one class at the first epoch while others have had good precision from the beginning.

Figure 5 shows the effect of the document length ($\kappa$: maximum number of sequences per document) on the precision of each model. LSTM-based models tend to perform quite well with not-so-long documents, but according we increase the input's length, the performance falls down. In contrast, concatenation-based models (5 and 6), perform better with long documents, even with sequences of 20, the performance increases.

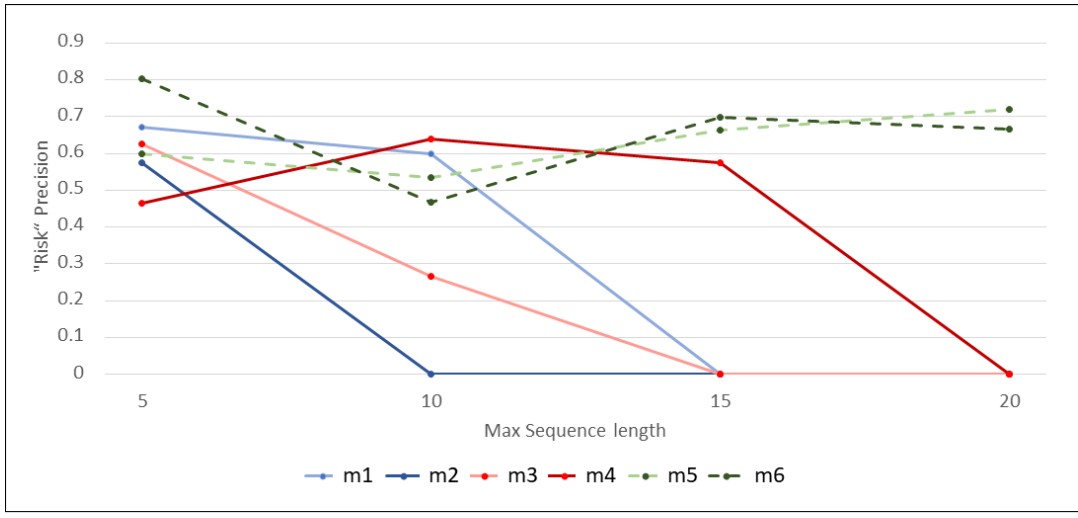

Figure 5: Comparison of Results of the models with respect to the maximum sequence length. If we increase the length, we will have models that can process longer documents.

## 6 DISCUSSION

As we showed in the results, even having an unbalanced dataset, model 4 demonstrates being the best model having the highest precision (73.30%). This model uses the most complex network configuration having a dense layer after the main components of the model. Compared with other studies that use numerical information instead of text, the results are not outperforming but are not far from those.

The different model's combination is the method that we consider important to do ablation studies. We can see that LSTM layers have a high impact on the model's performance for not-so-long documents, but do not perform very well with very-long documents. For this kind of model, reducing the dimensionality of BERT's outputs before integrating them into an LSTM model, is more convenient.

On the other hand, if we are working with a very long document, we should replace the LSTM network with a sequence of dense layers. The best result for this approach is obtained if we concatenate directly the BERT's output to the concatenation layer.

These results are very promising in terms of the use of pre-trained models like BERT together with an integration layer to process long sequences of texts. Even if the data is unbalanced, most of the models can make predictions of both classes since the first epoch.

## 7 CONCLUSION AND NEXT STEPS

Even if the default BERT models can not be used for processing long text, we can divide the document into segments and then process the document in parallel. For our case, this allows us to take advantage of the text in the Annual Accounts to determine if is possible to predict the level of a company going bankrupt based on unstructured data. The best precision of those models was $\approx$ 73%.

The experiments demonstrate that for having not-so-long documents, LSTM-based model performs better. On the other hand, if we need to process very-long documents, concatenation-based models are preferred.

In this sense, the text in the Annual Accounts' annexes has implicit risk information that can be extracted by a Deep Learning model. Our proposed model can be used in other domains which require processing long text.

We expect that our fine-tuned model has enough risk information that we are going to able to apply to other sources of financial text to predict the risk of bankruptcy.

The next step in our research is to combine the results of text models with numerical financial information. In addition, we are working on applying quantization and LORA methods to our BERT proposition to increase the complexity and quality of multi-modal architectures.

## ACKNOWLEDGEMENTS

**hidden for blind review**

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

## A APPENDIX DATASET

Figure 6 shows the distribution of the dataset with respect to the industries (obtained from the industry code of the company in LBR).

A detailed analysis of the number of documents and pages per language is contained in Table 5. Most of the documents' pages in the LBR dataset are in French ($\approx 81\%$) and the rest are in German ($\approx 8\%$) and English ($\approx 11\%$).

Table 5: Dataset distribution per language.

| Language | $\Sigma$ documents (d) | $\Sigma$ pages (p) | $\mu$ p |
|---|---|---|---|
| French | 16,796 (83.98% ) | 120,187 (80.79%) | 7.15 |
| German | 1,815 (9.07% ) | 12,215 (8.22%) | 6.73 |
| English | 1,389 (6.95% ) | 16,348 (10.99%) | 11.76 |
| Total | 20,000 | 148,750 | 7.43 |

After analyzing the tokenization of each document, we place the results in Table 6.

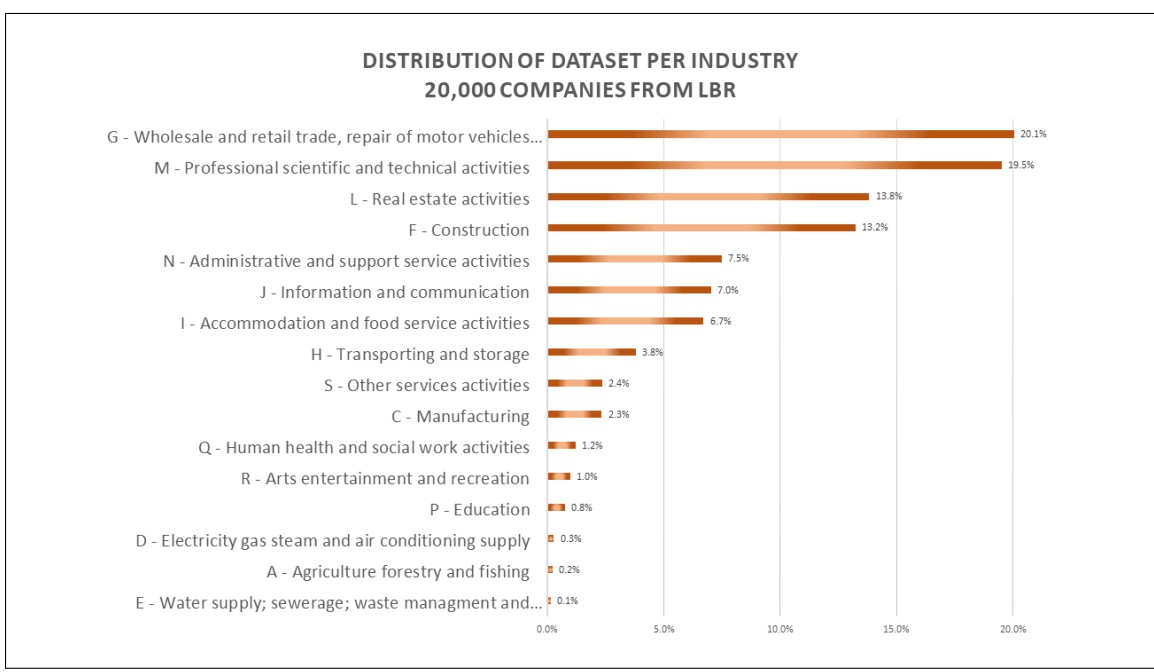

Figure 6: Distribution of companies per industries in the dataset

Table 6: Tokenization distribution per language.

| Language | Σ words (w) | Σ tokens (t) | μ t/w | μ w/d | μ t/d |
|---|---|---|---|---|---|
| French | 26.6M (82.1%) | 33.3M (82.6%) | 1.25 | 1,585 | 1,984 |
| German | 2.0M (6.2%) | 3.1M (7.8%) | 1.55 | 1,104 | 1,731 |
| English | 3.8M (11.7%) | 3.9M (9.6%) | 1.02 | 2,756 | 2,819 |
| Total | 32.4M | 40.3M | 1.24 | 1,623 | 2,019 |

Is interesting to notice from Table 5, that there are more documents in German than in English. In contrast, documents in German have fewer pages. Also, the language that generates fewer tokens per word, as shown in Table 6 is English (1.02), which is almost one token per word on average. On the contrary, German generates more tokens per word (1.55).

Table 5 also shows how many words and tokens per document on average there are. As we can see if we have a model with a maximum input size of 512 tokens, we will need on average 4-5 BERT instances per document. But using this average we will not be able to process longer documents properly.

We consider a company as bankrupted and we label them as *Risky* if it has a bankruptcy court order or a modification report that contains in the description the words *faillite* or if the company has a deletion date in its LBR profile webpage. The documents which have a modification report with the word *Voluntary* or a legal observation with the words *insolvabilité* or *faillite*, have been removed from the scope because it can not be determined if the company is going to be closed because of bankruptcy, voluntary liquidation or probable insolvency. The remaining companies are labeled as *Not Risky*.

Table 7 shows the distribution of the labels with respect to each language. Is clear the dataset is imbalanced with respect to the risk level, where around $\approx 20\%$ of the data is *Risky* and the rest is *Not Risky*. The data is imbalanced as well at the language level.

For training we are using 70% of the data, randomly selected but keeping the same imbalance ratio per risk level(Table 8).

Table 7: Distribution of Risk labels per language.

| Language | # Risky | # Not Risky | Total |
|---|---|---|---|
| French | 3,371 (88.0%) | 13,425 (83.0%) | 16,796 (84.0%) |
| German | 322 (8.5%) | 1,493 (9.2%) | 1,815 (9.1%) |
| English | 136 (3.5%) | 1,253 (7.8%) | 1,389 (6.9%) |
| Total | 3,829 (19.1%) | 16,171(80.9%) | 20,000(100%) |

Table 8: Distribution of Risk labels in train and validation.

| Dataset type | # Risky | # Not Risky | Total |
|---|---|---|---|
| Train | 2,680 (69.9%) | 11,319 (69.9%) | 13,999 (69.9%) |
| Validation | 1,149 (30.1%) | 4,852 (30.1%) | 6,001 (30.1%) |
| Total | 3,829 (19.1%) | 16,171(80.9%) | 20, 000 (100%) |

# A APPENDIX MODEL PROPOSITION

## A.1 II. SEGMENT ANALYSIS

As explained before and shown in Figure 1, each segment is fed into the BERT model. The BERT model by definition uses the first encoded layers to analyze the words in their context and then the last layers and pooler layer to work in the downstream task. We analyzed two different approaches for this phase. (a) To have a direct connection to the integration phase and (b) to include after BERT, dense layers to reduce the BERT's output dimension (768).

For the first approach (Figure 7), we just connect the last hidden state of the *[CLS]* token to the next phase. The last hidden state of *[CLS]* represents the whole segment in a vector of 768 features.

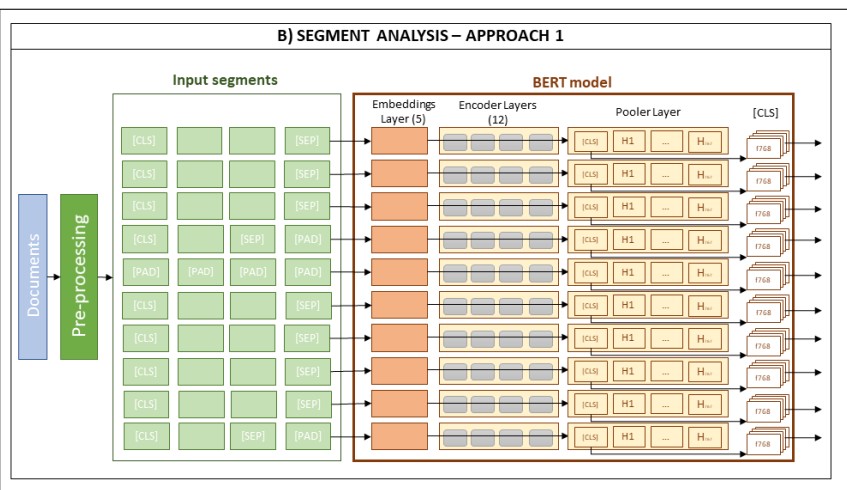

Figure 7: First approach of the segment analysis phase, we just output the last hidden state of the *[CLS]* token. This token represents the whole input segment in a vector of 768.

For the second approach, as shown by Figure 8, instead of delivering directly a summary vector of 768 features, we put two dense layers of size $\lambda_1$ and $\lambda_2$. We expect to reduce the dimensionality of the BERT output layer with two summarization layers, where each dense layer is smaller than the previous.

For this phase, we propose two different approaches. In the two first phases, we use an LSTM network for segment integration, ending with the classification layer. For the second approach, we reduce the LSTM's output dimensionality with intermediate dense layers. The third approach replaces the LSTM network with a concatenation layer and then a set of dense layers.

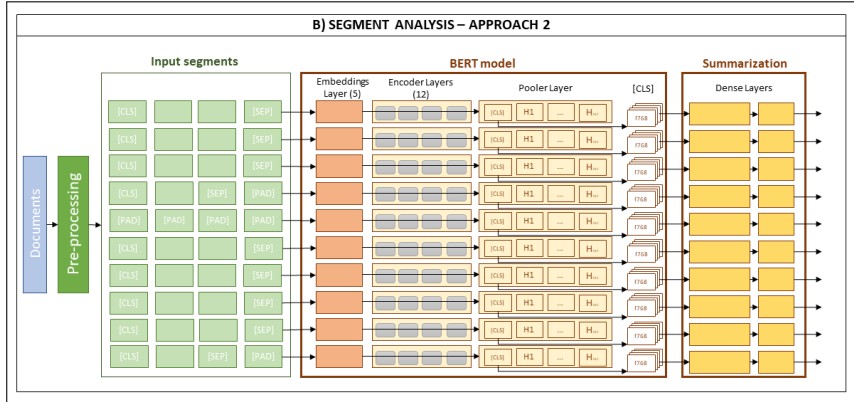

Figure 8: Second approach of the segment analysis phase, we connect the last hidden state of the *[CLS]* token with two consecutive dense layers. The main goal of this approach is to reduce the dimensionality of the network summarizing the BERT output.

## A.2  II. INTEGRATED PREDICTION

In the first approach of the integrated prediction phase (Figure 9), we use an LSTM to analyze the different segments of the document as a sequence. We only select the last hidden state of the last LSTM layer to feed the next dense layer. The input sequence length is determined by the maximum number of segments per document ($\kappa$). The number of LSTM hidden layers is defined for $\phi$. We are going to try different values to fine-tune this parameter.

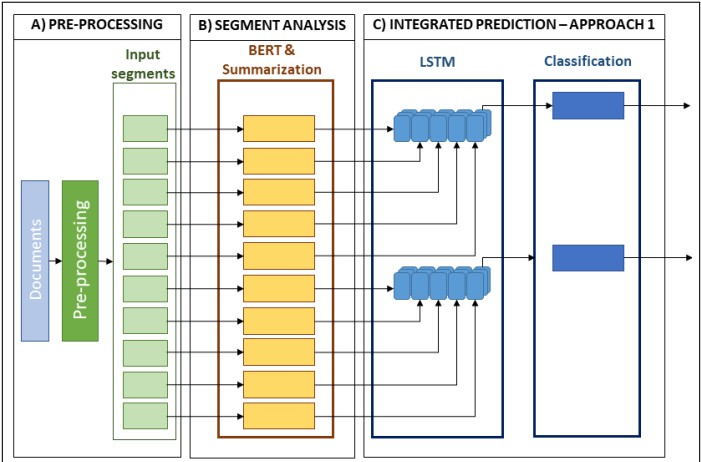

Figure 9: During the last phase we reconstruct the whole document integrating the different segments from any of the previous approaches (Direct BERT output or summarized output).

The difference between the second approach with respect to the first one is that we add intermediate dense layers to reduce the dimensions of the LSTM output, before the classification layer. We call these summarization layers (Figure 10).

The third and last approach for the integration and prediction phase, we replace the LSTM network with a dense layer that concatenates the outputs of the segment phase (Figure 11). We add summarization dense layers to reduce the number of dimensions of the concatenation dense layer and connect with the classification layer.

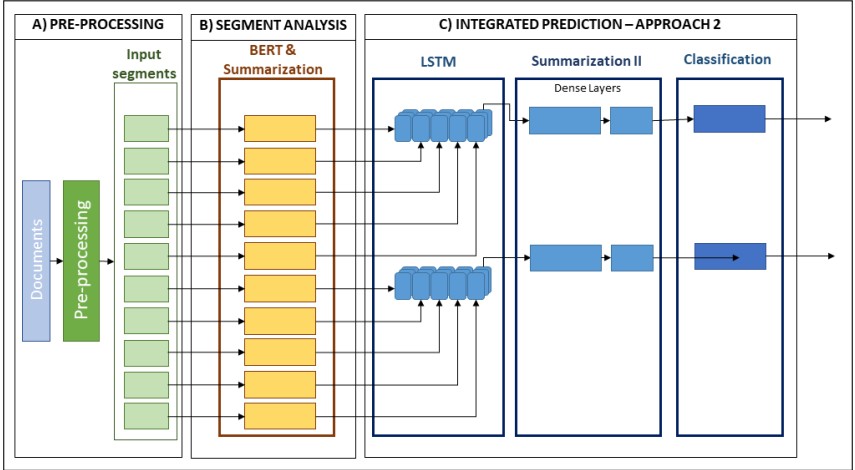

Figure 10: During the last phase we reconstruct the whole document integrating the different segments from any of the previous approaches (Direct BERT output or summarized output).

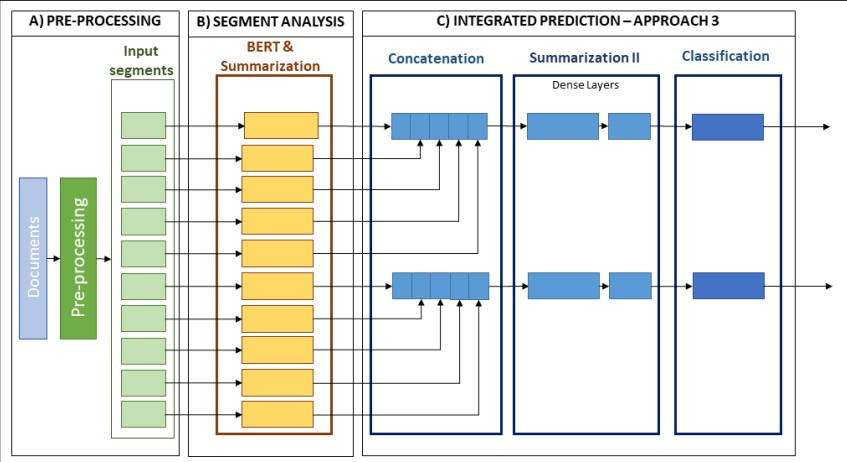

Figure 11: During the last phase we reconstruct the whole document integrating the different segments from any of the previous approaches (Direct BERT output or summarized output).

# B APPENDIX DETAILED RESULTS

Table 9 shows the results of the execution of the different models changing the number of hidden layers of the LSTM network. In the case of the models m5 and m6, as these do not have an LSTM network, we are reporting only the base execution.

Table 10 shows the results of the execution of the different models changing the number of segments (maximum input text allowed).

Table 9: Results of execution of the different test cases

| Test Case | Accuracy | F1 score | Precision | Recall |
|---|---|---|---|---|
| $tc\_m_1\_h_{30}$ | 0.8991 | 0.8876 | 0.5187 | 0.9197 |
| $tc\_m_1\_h_{50}$ | 0.8990 | 0.8875 | 0.5204 | 0.9157 |
| $tc\_m_1\_h_{80}$ | 0.9253 | 0.9217 | 0.5126 | 0.9089 |
| $tc\_m_1\_h_{120}$ | 0.8898 | 0.8747 | 0.4638 | 0.9221 |
| $tc\_m_1\_h_{200}$ | 0.9365 | 0.9327 | 0.6427 | 0.8496 |
| $tc\_m_2\_h_{30}$ | 0.8110 | 0.7291 | 0.0139 | 0.9411 |
| $tc\_m_2\_h_{50}$ | 0.8113 | 0.7302 | 0.0165 | 0.9047 |
| $tc\_m_2\_h_{80}$ | 0.8203 | 0.7510 | 0.0652 | 0.9493 |
| $tc\_m_2\_h_{120}$ | 0.9123 | 0.9054 | 0.6083 | 0.9019 |
| $tc\_m_2\_h_{200}$ | 0.919 | 0.9138 | 0.6510 | 0.8979 |
| $tc\_m_3\_h_{30}$ | 0.9091 | 0.9014 | 0.5892 | 0.9026 |
| $tc\_m_3\_h_{50}$ | 0.9145 | 0.9081 | 0.6205 | 0.9025 |
| $tc\_m_3\_h_{80}$ | 0.9113 | 0.9040 | 0.6171 | 0.9043 |
| $tc\_m_3\_h_{120}$ | 0.9191 | 0.9144 | 0.5256 | 0.9110 |
| $tc\_m_3\_h_{200}$ | 0.9168 | 0.9113 | 0.6405 | 0.8953 |
| $tc\_m_4\_h_{30}$ | 0.9073 | 0.8994 | 0.5839 | 0.8958 |
| $tc\_m_4\_h_{50}$ | 0.9020 | 0.8927 | 0.5552 | 0.8923 |
| $tc\_m_4\_h_{80}$ | 0.8085 | 0.7229 | 0.7330 | 0.8134 |
| $tc\_m_4\_h_{120}$ | 0.8901 | 0.8751 | 0.4734 | 0.9204 |
| $tc\_m_4\_h_{200}$ | 0.8715 | 0.8498 | 0.5230 | 0.9037 |
| $tc\_m_5$ | 0.9141 | 0.9071 | 0.6021 | 0.9044 |
| $tc\_m_6$ | 0.8931 | 0.8795 | 0.4856 | 0.9177 |

Table 10: Results of execution of different sequence lengths

| Test Case | Accuracy | F1 score | Precision | Recall |
|---|---|---|---|---|
| $tc\_m_1\_s_5$ | 0.9076 | 0.9027 | 0.6706 | 0.8893 |
| $tc\_m_1\_s_{10}$ | 0.9261 | 0.9215 | 0.5995 | 0.8029 |
| $tc\_m_1\_s_{15}$ | 0.7763 | 0.6786 | 0.0007 | 1.0000 |
| $tc\_m_1\_s_{20}$ | 0.7763 | 0.6786 | 0.0007 | 1.0000 |
| $tc\_m_2\_s_5$ | 0.8926 | 0.8832 | 0.5752 | 0.9125 |
| $tc\_m_2\_s_{10}$ | 0.7763 | 0.6786 | 0.0007 | 1.000 |
| $tc\_m_2\_s_{15}$ | 0.7763 | 0.6786 | 0.0007 | 1.000 |
| $tc\_m_2\_s_{20}$ | 0.7763 | 0.6786 | 0.0007 | 1.000 |
| $tc\_m_3\_s_5$ | 0.9036 | 0.8966 | 0.6244 | 0.9188 |
| $tc\_m_3\_s_{10}$ | 0.8543 | 0.8201 | 0.2663 | 0.9080 |
| $tc\_m_3\_s_{15}$ | 0.7763 | 0.6786 | 0.0007 | 1.000 |
| $tc\_m_3\_s_{20}$ | 0.7763 | 0.6786 | 0.0007 | 1.000 |
| $tc\_m_4\_s_5$ | 0.8713 | 0.8545 | 0.4657 | 0.9191 |
| $tc\_m_4\_s_{10}$ | 0.9295 | 0.9262 | 0.6402 | 0.7969 |
| $tc\_m_4\_s_{15}$ | 0.8901 | 0.8808 | 0.5745 | 0.8975 |
| $tc\_m_4\_s_{20}$ | 0.7763 | 0.6786 | 0.0007 | 1.0000 |
| $tc\_m_5\_s_5$ | 0.8983 | 0.8900 | 0.5991 | 0.9178 |
| $tc\_m_5\_s_{10}$ | 0.9038 | 0.8934 | 0.5340 | 0.9127 |
| $tc\_m_5\_s_{15}$ | 0.9083 | 0.9030 | 0.6624 | 0.9016 |
| $tc\_m_5\_s_{20}$ | 0.9150 | 0.9119 | 0.7205 | 0.8774 |
| $tc\_m_6\_s_5$ | 0.9308 | 0.9296 | 0.8017 | 0.8783 |
| $tc\_m_6\_s_{10}$ | 0.8901 | 0.8753 | 0.4682 | 0.9180 |
| $tc\_m_6\_s_{15}$ | 0.9141 | 0.9101 | 0.6982 | 0.8949 |
| $tc\_m_6\_s_{20}$ | 0.9090 | 0.9038 | 0.6661 | 0.9012 |

