# OpenReview forum: "Long BERT for bankruptcy prediction"
_ICLR.cc/2024/Conference — Submitted to ICLR 2024_

### Official Review · Reviewer_h86L · 2023-10-22

**Soundness:** 1 poor
**Presentation:** 3 good
**Contribution:** 1 poor
**Rating:** 3
**Confidence:** 4

**Summary:**

The research discusses the limitations of current bankruptcy risk prediction models, which primarily rely on numerical data. These models don't harness the available textual information. The study introduces a BERT-based model that predicts bankruptcy risk using long text sequences from Annual Accounts' annexes, even in the absence of explicit risk information. The researchers processed these texts in parallel and integrated them for predictions. They used a dataset of 20,000 annexes from Luxembourg's non-financial companies to train and validate the model, achieving a validation precision of around 73%. The model effectively learns risk information from unstructured and diverse long textual data.

**Strengths:**

1. This paper proposes a method for predicting bankruptcy risk using long text.
2. The author implemented the method and conducted experiments using the dataset of the Annual Accounts of non-financial companies from Luxembourg, and the accuracy rate exceeded 70%.

**Weaknesses:**

1.	The method proposed in the paper lacks novelty. As shown in Table 7, the risk prediction task can essentially be framed as a text classification problem (Risky or Not Risky). However, it's worth noting that similar research utilizing a combination of BERT and LSTM for long text classification was introduced four years ago [1].
2.	A notable shortcoming of the paper is the absence of comparative experiments. The author did not provide a comparison between their proposed method and established techniques. Furthermore, there was no evaluation against advanced language models such as GPT-4 and LLaMa, which are capable of handling a wide range of tasks.
3.	The paper also suffers from limited references to recent research. In addition to sparse citations of web pages and open-source code, the paper only includes references to a single study from 2022 and one from 2023. This lack of engagement with the latest research trends and findings weakens the paper's overall scholarly depth.

[1] Ashutosh Adhikari, Achyudh Ram, Raphael Tang, Jimmy Lin: DocBERT: BERT for Document Classification. CoRR abs/1904.08398 (2019).

**Questions:**

1.	Can the long text classification method mentioned in the "Weaknesses" section effectively address the issue of bankruptcy risk prediction when trained on the dataset presented in this paper?
2.	What are the improvements and advantages of the method in the paper when compared to DocBERT and its latest improved methods?
3.	In comparison to large language models combined with in-context learning methods, does the method proposed in the paper demonstrate an accuracy advantage?

---

> ### Author Response · Authors · 2023-11-20
> **Reply to comments**
>
> Thanks to the reviewer for the feedback.
>
> Here we are addressing the comments:
> - The research paper recommended DocBERT, it might lead to confusion if is not read carefully. Besides the name is DocBERT, the authors propose the distillation of BERT into an LSTM model, that means that the final model used for prediction is an LSTM model. As is well known, LSTM models were replaced by BERT mainly because is unable to process in parallel and also not good in very long sequences (Of course are better than the RNN). This was reflected in our results, more than 5 LSTM time sequences, the performance drops. DocBERT uses a dataset that according to the provided dataset table, the maximum average number of words per document is bellow 400. Considering only 1.2 tokens per word, we are dealing with a maximum number of 480 tokens, even less than one BERT segment. Our dataset in average has 2.8K tokens per document and 2.7K words per document (English). This is the reason that a pure LSTM or BERT network does not work for processing long text.
> - Also as explained in the paper, we compared with established techniques for bankruptcy prediction (that are mainly numbers) and the text itself is not better than that. Because of this reason our project main goal is to have an integrated model that is having text information, numerical information and  other sources of information.
> - GPT models like Llama 2  and GPT-3 are broadly used nowadays, but the main problem of these family of models (generative) is the hallucination. We are using also these models to process the text, but the amount of dirty answer that we receive is above 30%. Commercial models like GPT-4 and even the best performed versions of GPT3.5 can be costly for some companies that needs to process millions of documents. For this reason, we are focused on ways to have lighter classification models like BERT-Based models, faster and cheaper to implement. Nevertheless, we are including in the paper the results of using Llama 2 for bankruptcy prediction and the current limitations.
> - The references provided are the ones that are focused on BERT, lately most of the research done was following the wave of GPT-based models, but those are not the models that we are pursuing because of the reason explained before. There is not much work related to bankruptcy prediction using long text sequences (or in other domains), and this is why we considered important to share our results and contributions.

---

### Official Review · Reviewer_8GNw · 2023-10-30

**Soundness:** 3 good
**Presentation:** 3 good
**Contribution:** 3 good
**Rating:** 5
**Confidence:** 3

**Summary:**

The article discusses a novel approach to predicting bankruptcy risk for companies using textual information from the annexes of annual accounts. This is in contrast to most existing bankruptcy prediction models, which rely on numerical data like financial statements and ratios.

The authors propose a BERT-based model that processes parallel segments of a document with BERT and then integrates them for a unified prediction. They use a dataset of 20,000 annexes from the Annual Accounts of non-financial companies in Luxembourg to train and validate their model. Their results show that two models achieved a validation precision of approximately 73%, with performance varying based on the length of the documents.

The study highlights the potential of using deep learning models to extract risk information from unstructured textual data and suggests that this approach can be applied to other domains requiring the analysis of long text.

**Strengths:**

- The article makes a contribution to a lightly researched topic. Bankruptcy prediction using textual data is often focussed on news sentiment, while the authors target the annual statements by the company.
- The authors contribute to the literature on using arbitrary length textual data in machine learning applications.

**Weaknesses:**

- It is unclear what the main contribution is; is it the application of an arbitrary length BERT model to a specific dataset or is it the specific formulation of the arbitrary length BERT model.
- While it is nice that the authors use the textual information from the appendices to predict imminent bankruptcy, it is not particularly meaningful given that the predictive information in there is not any better than the financial ratios from the tabular data in those same reports. It would mostly be interesting whether the combined predictive value exceeds the predictive value from the ratios alone.

**Questions:**

- Was there a specific reason for how you built your arbitrary length BERT model as opposed to different versions in your references? Does your version perform better for general text classification tasks or did you just find it to work better for the bankruptcy prediction task?
- Can you explain how the different languages are processed by your model? Does the processing to tokens effectively remove the differences?
- Is there a way to determine some feature importance from your model. Could there be standard phrases in the dataset suggesting imminent bankruptcy that suggest that it should be relatively easy for a machine learning model to predict a company going under?

---

> ### Author Response · Authors · 2023-11-20
> **Reply to the comments**
>
> Thanks to the reviewer for the feedback.
>
> Here we are addressing the comments:
> - We are clarifying that besides the research object of bankruptcy prediction we are also contributing to the architecture for dealing with long text using BERT models that can run even in a laptop with 1 GPU. We propose to use BERT+LSTM architectures for not so big documents (<=5 segments) and use BERT+concatenation for bigger documents (more than 5).
> - With respect to the question 1, We were applying these architectures in other models where we need to feed by page (3 segments) and other long documents (8 segments). The accuracy for other classification tasks we had were around 97 % (not related to bankruptcy). We are planning to create another paper to focus on these different use cases that the architecture can allow us.
> - With respect to the question 2, the model BERT itself despite the fact of handling multiple languages, those embeddings are not fully related between each other, it can be seemed like parallel models per language. We could easily prove by comparing the distance between the embeddings of one word in different languages. In the other hand the more complexity of the GPT models can handle this problem easier. You can provide a text with different languages and to expect the answer in a specific language. The main problem of this GPT for classification is that there are going to hallucinate and create answer that could be difficult to process unless you apply more layers of validation, and also the resources required are bigger. One of our open research questions is how to address BERT embeddings of different languages to work together.
> - With respect to the question 3. Thanks for the contribution, we are starting to include Explainability and ranking the top tokens that can lead us to the bankruptcy prediction.

---

> > ### Comment · Reviewer_8GNw · 2023-11-22
> >
> > Thank you for your comments. I will maintain my score.

---

### Official Review · Reviewer_WCJF · 2023-11-04

**Soundness:** 2 fair
**Presentation:** 3 good
**Contribution:** 2 fair
**Rating:** 3
**Confidence:** 3

**Summary:**

The authors consider the problem of predicting whether a company is at risk of bankruptcy using modern natural language processing (NLP) and deep learning techniques.  In particular, the authors propose a methodology that first divides large text documents into multiple segments, each of which are evaluated independently using the BERT model, and then combines the output of the BERT model on each segment so as to produce a final Risk or Non Risky prediction.  A variety of compression and integration schemes are considered and the authors demonstrate performance that while not as strong as purely numeric methods, does a reasonable job. The main technical contribution appears to be a methodology for applying the BERT model to large text documents that have a number of tokens that exceed BERT's input dimension.

**Strengths:**

Overall, I thought this was an interesting and fairly well executed empirical paper.  The authors clearly identified the problem to be solved - predicting bankruptcy risk using only text data - and motivate its importance using an actual corpus of financial statements from Luxemborg.  The authors then describe the key challenge one faces when applying an off the shelf language model, like BERT, to large financial documents and nicely walk through their technical approach to solving this problem.  Finally, they evaluate the performance of several variants on the Luxemborg financial statements data set, demonstrating solid performance for a purely text based model.

I am not an expert in the application of BERT models and am unfamiliar with the common heuristics one might use to apply BERT to long documents, but this paper does present a reasonable approach.

**Weaknesses:**

My main criticism of this paper is in the empirical findings.  First, it was rather unclear how the training and test set were determined. Was a random selection of documents held out in the test set?  Was the test set the same as that used by Mai et al.?  Finally, where do the "Risky" vs "Non Risky" labels come from?

Next, I can accept the fact that the performance of a pure text based classifier might not be as good as one that uses financially relevant numeric values, but I think the authors should have directly compared their performance wrt Mai et al.'s work or even a basic approach, e.g. randomly sampling a single 512 token segment from each training document.  Doing so would have demonstrated the benefit of the summarization and integrated prediction approach and justified their usage.

I also found the tables and figures in the experiments and results section hard to interpret.  Was Table 3 an OOS performance measure? How does recall vary by each model configuration? And what conclusion am I supposed to draw from Figure 4? How does it differ from table 3?

Last, I question whether the technical contribution here is significant enough for publication at ICLR.  While I know that a lot of time, energy and technical expertise was required to generate the empirical findings, I'm not sure how broadly this work can be applied.

**Questions:**

-what does the following phrase on page 4 mean: "For this reason, the shuffling and selection of train, test and validation data are done before the pre-processing"?
-doesn't the number of segments (and therefore the number of documents trimmed) also depend on delta, the number of overlapping tokens?
-is sequence/overlap important at all?  What if one just randomly samples segments instead?
-Can this approach be applied to other corpuses with large text documents.

---

> ### Author Response · Authors · 2023-11-20
> **Reply to comments**
>
> Thanks to the reviewer for the feedback.
>
> Here we are addressing the comments:
> - The test set was determined randomly by document. As the model propose to use many segments for each document, we can not do randomization of the segments by default, we need to do it by document to avoid losing the integrity in the document, because it will allow us to integrate in the next phase in a LSTM or concatenation model. As explained in the paper, the weakness of the paper of Mai et al, as other similar research, is that they are using the risk section of the SEC fillings, where are implicitly mentioning the level of risk of the company and an analysis, in that case they do not need to process the entire document, just the paragraphs that contains that information. In our case the dataset does not contain an explicit section of the risk analysis, and this is why we need to feed the entire annexes in order to look for risk information. We thank the proposition of selecting randomly 512 tokens (510) of each document to use as a base line to compare our model.
> - Also, as we mentioned and explain, because of the high unbalanced data, we are just focusing on the precision of risk prediction as the performance measurement, because if we see the other metrics like recall or accuracy, the model is above 90%, that can lead us to bad conclusions. We thank the reviewer that we agree that information of the Figure 4 and Table 3 are the same and should be integrated.  As explained in the previous paragraph the conclusion of the figure 4 is that the best performed model for not so long text sequences is to have an LSTM of 80 hidden layers with the proposed model m4 (applying dense layers after BERT and after LSTM). Bigger than that, the model starts to be too complex and to lose performance.
> - As mentioned in the paper, the prediction of the bankruptcy is one of the goals, the other objective is to provide with two architectural options (LSTM or concatenation) for dealing with long text prediction. In our research project we are already using these models to do predictions by documents (other labels) and by pages, that requires more than 510 tokens. In consequence, the architecture can be broadly applied to any domain that requires the use of BERT models for long text prediction.
> - About the question of shuffling, is because we shuffle the data at document level because in the preprocessing stage, we already have the documents divided into segments. To avoid the corruption of the integrity of the documents itself, we do the shuffling before the pre-processing phase.
> - With respect to the number of segments kappa that depends on delta, we correct the equation 1, because as commented, the number of overlapping tokens affects the number of required segments. Is important because it allow the model to keep the previous context in each segment analysis. As suggested by the other reviewed we are going to use random selection of segments to make a base line to compare with.

---

### Official Review · Reviewer_QYDH · 2023-11-06

**Soundness:** 2 fair
**Presentation:** 2 fair
**Contribution:** 2 fair
**Rating:** 3
**Confidence:** 4

**Summary:**

The authors suggest using document segments analyzed via BERT, with and without LSTM networks, for bankruptcy prediction. Despite an unbalanced dataset, their best model attained a 73.3% precision rate. The research also reveals that models using concatenation are nearly as precise as other models discussed in the paper and excel with longer texts.

**Strengths:**

- Capable of forecasting bankruptcy using extensive text sequences in multiple languages: French, German, and English.
- Includes an analysis of the model's performance in relation to document length and variations in the number of hidden layers.

**Weaknesses:**

- Relies solely on textual information, and as acknowledged by the authors, numerical data remains crucial for risk prediction.
- Should still compare to traditional default prediction models in finance, like FIM
- It will be practical to predict a term structure of bankruptcy probabilities, not just treat it as a simple classification task.

**Questions:**

none

---

> ### Author Response · Authors · 2023-11-20
> **Reply to the comments**
>
> Thanks for the feedback.
>
> Here we are addressing the comments:
> - One of the main goals of this paper is to see how good a bankruptcy prediction can be done with a model using only textual information, and it was mentioned in the abstract and conclusions, this is part of a bigger project that is going to integrate textual information, numerical information and other sources of information. We are going to put more emphasis in this in the paper to make it clearer.
> - We have compared the results of our model with other methods like Logistic Regression, Random Forest and LightBM, that are also working with a very similar data. We thank the reviewer for providing the info about the FIM models but, we could not find a specific finance paper about a method / model FIM for bankruptcy prediction. We thank in advance is the paper's name can be provided in order to improve our work.
> - To provide a structure of probabilities as a result of the model is a very interesting proposition, we have started working in doing this based on the classification result and three classes according to the degree of confidence, to still be able to use the current labeling.

---

### Meta-Review · Area_Chair_EeeD · 2023-12-14

**Metareview:**

All reviewers agree that this paper unfortunately should be rejected. It does not make a significant contribution to ML research. As one reviewer wrote, "The main technical contribution appears to be a methodology for applying the BERT model to large text documents that have a number of tokens that exceed BERT's input dimension." That input dimension is about 500, which is far behind the state of the art nowadays for LLMs.

From a business perspective, the approach has at least three significant weaknesses, in addition to those highlighted by reviewers. First, the dataset should be split by time, not by random shuffle. For example, data from 2022 should be the test set, 2021 validation, and earlier for training. In finance, non-stationarity is ubiquitous and makes random data splits lead to over-optimism. Second, the authors should report results for a modern LLM "off the shelf." Third, the dataset is not very unbalanced: 20% of companies are labeled risky. True bankruptcy is much more rare. The paper should explain why it is truly useful to classify this notion of "risky." What, if anything, is actuallt being predicted?

**Justification For Why Not Higher Score:**

No significant contribution to ML or to finance.

**Justification For Why Not Lower Score:**

Somewhat interesting.

---

### Decision · Program_Chairs · 2024-01-16

Reject